# Bilateral Germ Cell Tumor of the Testis: Biological and Clinical Implications for a Stem Versus Genetic Origin of Cancers

**DOI:** 10.3390/cells14090658

**Published:** 2025-04-30

**Authors:** Jamaal C. Jackson, Darren Sanchez, Aron Y. Joon, Marcos R. Estecio, Andrew C. Johns, Amishi Y. Shah, Matthew Campbell, John F. Ward, Louis L. Pisters, Charles C. Guo, Miao Zhang, Niki M. Zacharias, Shi-Ming Tu

**Affiliations:** 1Department of Urology, The University of Texas MD Anderson Cancer Center, Houston, TX 77030, USA; jjackson10@mdanderson.org (J.C.J.); dsanchez7@mdanderson.org (D.S.); jfward@mdanderson.org (J.F.W.); lpisters@mdanderson.org (L.L.P.); 2Department of Biostatistics, The University of Texas MD Anderson Cancer Center, Houston, TX 77030, USA; ayjoon@mdanderson.org; 3Department of Epigenetics and Molecular Carcinogenesis, The University of Texas MD Anderson Cancer Center, Houston, TX 77030, USA; mestecio@mdanderson.org; 4Department of Genitourinary Medical Oncology, The University of Texas MD Anderson Cancer Center, Houston, TX 77030, USA; acjohns@mdanderson.org (A.C.J.); ayshah@mdanderson.org (A.Y.S.); mcampbell3@mdanderson.org (M.C.); 5Department of Pathology, The University of Texas MD Anderson Cancer Center, Houston, TX 77030, USA; ccguo@mdanderson.org (C.C.G.); mzhang8@mdanderson.org (M.Z.); 6Division of Hematology/Oncology, University of Arkansas for Medical Sciences, Little Rock, AR 72205, USA

**Keywords:** bilateral testicular cancer, cancer stem cells, origin of cancers, tumor heterogeneity

## Abstract

Germ cell tumors of the testis (GCTs) provide an ideal tumor model to investigate the cellular versus genetic origin of cancers. In this single institutional study, we evaluated 38 patients with bilateral GCT, including tumors that occurred simultaneously (synchronous) and those occurring at different times (metachronous). For nine of these patients, DNA was isolated from the right and left GCT to determine the genomic and epigenetic differences between tissues using whole-exome sequencing (WES) and reduced representation bisulfite sequencing (RRBS). We found that seminomas and non-seminomas are molecularly distinct based on DNA methylation and not due to synchronous or metachronous disease. In addition, we did not observe conservation of genetic mutations in right and left GCT in either synchronous or metachronous disease. Our data suggest a cellular origin for bilateral GCT.

## 1. Introduction

Currently, there is ongoing debate regarding whether cellular context (i.e., stem-ness origin) or genetic content (i.e., genomic aberrations) dictates the malignant predilection and clinical manifestation of various tumor phenotypes. Clarification as to whether there is a cellular versus genetic origin of cancer would affect our understanding of the heterogeneity of cancer and our design of multimodal strategies to treat it.

Germ cell tumors of the testis (GCTs) provide us with an ideal tumor model to investigate the cellular versus genetic origin of cancers. Since germ cells are the origin of all fetal cells, a germ cell is a prototype stem cell and GCT is the archetype of stem cell tumors with relatively few but putatively pivotal genetic defects (e.g., i [12p]). GCT’s intra- and intertumoral heterogeneity is clear-cut. Importantly, it is one of the few curable solid malignancies, even in cases of advanced disease.

In addition, the ability to access a complete tumor in an intact organ, as is the case in patients who undergo radical orchiectomy, is invaluable for an investigation of its pathogenesis. The developmental pathway and migratory timeline of primordial germ cells (PGC) are well established [1,2,3,4]. A comparison of the epigenetics and genetics of disparate tumors in both synchronous or metachronous bilateral GCT could help to discern if the tumor’s origin is genetic or cellular.

Bilateral GCT occurs in <2% of patients with GCT [5]. After a primary GCT, the probability of developing a separate GCT in the contralateral testis increases about twelve-fold [6]. Notably, 92% of synchronous tumors have a seminoma rather than a nonseminoma histology. If the first tumor was observed to have a seminoma histology in the setting of bilateral GCT, regardless of the timing of incidence, the same histology was observed in 69% of second tumors. Furthermore, when the interval between metachronous tumors was <60 months, there was a seminoma histology in 66% of cases [6].

Studies of GCT have revealed that the mutational profiles are similar between seminomas and nonseminomas, with a markedly low rate of somatic mutation (mean rate of 0.50 and 0.49 somatic mutations/Mb, respectively) compared with lung cancer (8.0 mutations/Mb) and melanoma (11.0 mutations/Mb) [7,8]

The most frequently mutated GCT gene is KIT, in which activating mutations are observed in 25–30% of seminomas [9,10]. KRAS mutations occur with a frequency of 5–10%. To date, no one has discovered any concordant mutations in bilateral GCT, regardless of temporality (i.e., synchronous or metachronous) or histology (i.e., seminoma or nonseminoma), suggesting that no common precursor mutation occurs during oncogenesis [11].

Solving the controversy regarding the cellular versus genetic origin of cancer and how this is associated with intratumoral heterogeneity, metastatic potential, and drug resistance—has broad clinical implications for improving personalized care, as it could determine whether it is better to target the cellular context or the genetic defect. In this single institutional study, we evaluated 38 patients with bilateral GCT. For nine of these patients, DNA was isolated from FFPE (formalin-fixed paraffin-embedded) blocks of the right and left lesions to determine the similarities/differences between the two tissues in terms of their somatic mutations and DNA methylation.

## 2. Materials and Methods

A search of the database at MDACC was performed and patients with bilateral GCT were identified. The SlicerDicer function within EPIC was then utilized to generate a list of men who developed a bilateral nonseminomatous germ cell tumor (NSGCT) or seminoma seen at MDACC. The base population was designated as all patients evaluated at “Mays GU Onc” or “Mays Urology” between 28 February 1991 and 28 February 2021. This base was divided according to the “neoplasm by body system” (genital system, male) and ICD 10 codes for testicular cancer (C62.00, C62.01, C62.02, C62.10, C62.11, C62.12, C62.90, C62.91, C62.92), which narrowed the patient population to 1,695. This list was exported onto Excel for manual review. Duplicates were then removed, which decreased the patient list to 1205. This list was reviewed with reference to the inclusion criteria (bilateral testicular cancer, either seminoma or NSGCT, synchronous or metachronous), which initially yielded 28 patients.

The medical record was then reexamined for additional patients prior to the conversion to EPIC in 2016, whose information would have been entered into the chart as free text “historical conversion”, without the need for diagnosis codes and, therefore, missed when using the SlicerDicer method. An additional 10 patients were identified via personal communications (i.e., through multidisciplinary care discussions), leading to a total of 38 patients. This study excluded patients who received chemotherapy prior to either of their orchiectomies or whose orchiectomy showed a burnt-out tumor.

For categorical variables, Fisher’s exact test was used to determine the *p*-value between synchronous and metachronous disease. For continuous variables, the Mann–Whitney U test (also called the Mann–Whitney Wilcoxon Test) was used to determine the *p*-value between synchronous and metachronous disease. The log-rank *p*-value was determined between the overall Kaplan–Meir survival curves of synchronous and metachronous disease in our 38-patient cohort. The hazard ratio and median survival times between synchronous and metachronous disease were not calculated as one group had zero events.

### 2.1. Tumor DNA Isolation

FFPE blocks for right and left GCT were obtained from nine patients. For normal-tissue DNA, FFPE blocks from the same patient were used, who, according to their histology, showed no evidence of disease. These blocks were obtained via written consent on LAB02-152, which was reviewed and approved by the MD Anderson Institutional Review Board. H&E slides were cut and reviewed by a pathologist (MZ). Microdissection was performed for all samples. DNA was extracted using the Ionic FFPE to Pure DNA Kit with the Ionic Purification System (Purigen Biosystems, Pleasanton, CA, USA), which enables the automated purification of DNA from FFPE tissue samples.

### 2.2. Whole-Exome Sequencing (WES)

Eighteen genomic samples were sequenced using NovaSeq600 (Illumina, San Diego, CA, USA) using 150-base-pair end sequencing by Illumina with UMI. The FASTQ data from Illumina NovaSeq were processed at the Institute for Personalized Cancer Therapy (IPCT) at MD Anderson through the standard methods outlined in [12]. Briefly, after bcl files were demultiplexed to fastqs, they were aligned to human reference assembly hg19 using BWA-MEM (0.7.15/bwa mem -k 31 -T 100 -t 8 -M) [13]. This was followed by GATK mark_duplication, realignment, and recalibration to produce the final BAM files. MuTect v.1.1.4 was used to detect somatic mutations [14].

### 2.3. DNA Methylation Analysis

Library preparation was performed at the MDACC Epigenomics Profiling Core using the Ovation^®^ RRBS Methyl-Seq kit (Tecan AG, Männedorf, Switzerland). In brief, genomic DNA was digested with MspI, followed by adapter-ligation and repair. An illumina sequencing-compatible cytosine-methylated adaptor was ligated to the enzyme-digested DNA. Size-selected fragments representing sequences from approximately 40 bp to 400 bp were bisulfite-converted and library preparation was carried out via PCR amplification. Pooled libraries were sequenced on an Illumina NovaSeq 6000 instrument as 75 bp single reads to 60 million reads to reach an average 20-times coverage per CpG site. RRBS reads were aligned to a UCSC Genome Browser human reference genome using Bowtie and methylation was performed using Bismark v0.23.1 [15]. Differentially methylated CpG sites (DMCs) and differentially methylated regions (DMR) were identified using MethylKit v1.28.0 [16]. After filtering out CpG sites with fewer than 10 reads and those with the highest 0.1% coverage, we utilized the sites with the top 10% variation across samples to conduct unsupervised hierarchical clustering. Allowing for one sample to be absent in either group (R or L), we applied the hclust function from the dendextend package (version 1.17.1, at R version 4.1) to calculate the Euclidean distance between data points, using the “average” option as the linkage method to assess the distance between clusters. The highest value in the hierarchical cluster graph indicates the greatest separation between clusters. In addition, differential methylation analysis was performed to determine differences between seminomas (n = 11) and nonseminomas (n = 7) and between right (n = 9) and left (n = 9) tissues in the nine paired bilateral tumors. Methylation data were filtered to maintain the ≥10× coverage and to ensure sites were present in most samples, allowing for one sample to be missing from either group (S or NS; R or L). Genes were annotated using the hg19 genome [17]. Adjusted *p*-values between comparison groups were determined using the Benjamini–Hochberg procedure.

## 3. Results

### 3.1. Clinical and Pathological Features of Total Cohort

Orchiectomies are often performed before chemotherapy or radiation therapy. Patients are offered chemotherapy or radiation therapy prior to resection only in rare cases of high-volume symptomatic GCT. These patients were excluded from our study. Extensive staging was performed prior to any orchiectomy, which included an ultrasound of the contralateral testis; this confirmed that, in metachronous disease, the other GCT lesion was not present by imaging at the time of the initial orchiectomy.

We identified 38 patients with bilateral GCT who underwent their first orchiectomy between 1 January 1984 and 21 April 2022 and their second between 28 August 1997 and 21 April 2022. Median follow-up was 134.2 months (IQR 69.5–222.1 months). Median age at the time of the initial orchiectomy was 27.5 years (IQR 20–32 years), while median age at the time of the second orchiectomy was 33 years (IQR 26–37 years). Seven patients had synchronous tumors, while thirty-one had metachronous bilateral GCT. There were 13 bilateral seminomas, 12 bilateral nonseminomas, and 13 bilateral seminoma and nonseminoma (Table 1). For those patients with metachronous bilateral GCT, the median time between the two GCT occurrences was 47.7 months (IQR 19.6–108.9). We found no difference in the overall survival between patients with synchronous versus metachronous bilateral GCT (Appendix A). We found no statistical differences in all categorical variables (race, stage, pathologic subtype, disease progression, or survival percentage) between patients with synchronous versus metachronous disease; however, we did observe statistical significance in the median follow-up time. Because a period of time ranging from months to years can separate occurrences of contralateral GCT in the metachronous disease setting, the median follow-up time was three times longer for the metachronous cohort. Our results are similar to those obtained in a recent 1111-patient cohort with either synchronous or metachronous GCT, which found no differences in baseline clinical characteristics or a statistical difference in overall survival between groups (log-rank test *p*-value of 0.62) [18].

### 3.2. Clinical and Pathological Features of Nine Patients

We were able to obtain the left and right GCT from nine patients. Histological characteristics, interval between lesions, treatment, and pathology are shown in Table 2. The stage shown in the table is the stage at initial diagnosis. Four patients (SP2, SP4, SP5, and SP7) had synchronous tumors and five patients had metachronous disease (SP3, SP6, SP23, SP57, and SP74). The difference in months between tumors in the metachronous cohort ranged from 18.8 to 158.3 months. Patients were treated with either multiple rounds of BEP (bleomycin, etoposide, and cisplatin), VIP (etoposide, ifosfamide, and cisplatin), or external beam radiation (XRT). All patients were alive and disease-free at last follow-up.

### 3.3. Genomic Profiles

Whole-exome sequencing (WES) revealed that the occurrence of mutations was relatively infrequent in the 18 samples from nine patients with bilateral GCT. Out of the approximately 20,000 genes investigated, 189 (<1%) had a detectable mutation in the nine paired cases (n = 18). The mutations found in genes within each sample are illustrated in Figure 1A. A total of eight genes were mutated in more than one sample, including KIT (n = 4, 22%) and KRAS (n = 6, 33%) (Figure 1B). In patient SP2, the KIT mutation was concordant in the right and left synchronous seminomas, while in patient SP7, the KRAS mutation was concordant in the right and left synchronous seminomas. However, in patients SP4 and SP5, both KIT and KRAS mutations occurred in the right but not in the left synchronous seminomas.

### 3.4. Methylation Profiles

Methylated cytosines were mapped genome-wide with RRBS [19,20] in the 18 patient tissues. We observed a similar mapping efficiency of the left and right lesions in all samples (Appendix A); therefore, all nine pairs (18 samples) were used to determine the conservation of DNA methylation between the right and left GCT. Unsupervised hierarchical clusters were generated to provide visually distinguishable patterns between tissues (Figure 2). We found that the pairs of metachronous nonseminomas (i.e., patients SP3 and SP6) displayed a similar methylation profile but did not share any genetic mutations, suggesting a common cellular rather than genetic origin despite a separation in the time of their diagnoses of 21 and 67 months, respectively. On the other hand, even though both of patient SP2’s synchronous seminomas harbored KIT mutations, the methylation profile of his left-side seminoma more closely resembled that of patients SP23 and SP74’s right-side seminomas, which did not harbor any KIT mutation.

Intriguingly, both SP4R and SP5R were nonseminomas that harbored KIT and KRAS mutations, and they exhibited a similar methylation profile (i.e., clustered) suggesting a common genetic and cellular origin despite their presence in different patients. Because neither of their respective contralateral synchronous SP4L nor SP5L seminomas displayed any of these genetic or epigenetic signatures, we postulate that the initiating and promoting carcinogenic events within SP4R vs. SP4L and SP5R vs. SP5L must have occurred after the separation of their respective PGC in the gonadal ridge (i.e., after human embryonic day 33) [21]. In other words, their disparate genomic and methylation profiles suggest that they have separate cellular origins. It is conceivable that the initiating and promoting carcinogenic events in some synchronous GCTs occurred metachronously but were diagnosed synchronously. It is also plausible that SP5L seminoma may behave like a nonseminoma or a mixed nonseminoma with a predominant seminomatous component (i.e., an atypical seminoma with yolk sac tumor features).

### 3.5. Differential Methylation Comparison

We adopted a cutoff of a >40% mean difference in methylation between groups with adjusted *p*-values < 0.01 to identify differential methylation. We found 894 CpG sites that were hypermethylated in nonseminomas and none in seminoma tissues based on our criteria. The top 10, based on *p*-values, are shown in Table 3. To report on hypermethylated CpG sites in seminomas compared to nonseminomas, we used an adjusted *p*-value of 0.01 with >10% mean difference in methylation and found a single CpG site located in chromosome 7, position 148787751, gene ZNF786, with a mean difference of 10.8. A similar analysis was performed to determine the differential methylation between right and left GCT in the nine pairs with a >40% mean difference in methylation with an adjusted *p*-value of <0.01 and no genes were found.

For some of the highest methylation sites, the DNA region found to have differential methylation between seminoma and nonseminoma tissues encoded two different genes or two uncharacterized RNA genes (LOC101928401, LOC401357; LOC100507351, LOC100132174). Significantly higher DNA methylation was found in the nonseminomas in at least one gene site: *ARC* (apoptosis-repressor and the caspase recruitment domain), JRK (Jerky Homolog), *MATK* (megakaryocyte-associated tyrosine kinase), *ADARB2* (double-stranded RNA adenosine deaminase), *RAP1GAP2* (RAP1 GTPase-activating protein 2), *CRHR2* (corticotropin-releasing hormone receptor 2), *INMT* (indolethylamine N-methyltransferase), and *GALNT9* (polypeptide N-acetylgalactosaminyltransferase 9). The *TPPP1* (tubulin polymerization-promoting protein 1) gene was found to have higher methylation in nonseminomas at two different sites.

In addition, we specifically looked at the CpG sites in the *AR* (androgen receptor), *MAGE-A4* (melanoma-associated antigen A4), and *JUP* (junction plakoglobin) genes. These genes were previously found to be associated with testicular cancer [22,23,24]. We found differential methylation in multiple sites of *MAGE-A4* and one site of *JUP* between seminoma and nonseminoma tissues (Table 4). *AR* methylation on CpG sites was not statistically significant between seminoma and nonseminoma tissues.

## 4. Discussion

In this single institutional study of 38 patients with bilateral GCT, 7 patients had synchronous tumors, while 31 had metachronous bilateral GCT. Within our cohort, there were 13 bilateral seminomas, 12 bilateral nonseminomas, and 13 bilateral seminoma and nonseminoma. Within our cohort, only two patients died of their disease, and we found that the overall survival was not statistically different between patients with synchronous and metachronous bilateral GCT.

We were able to obtain the left and right GCT tissue from nine patients. All nine patients were alive at last follow-up and had no evidence of disease. Four of these patients had synchronous bilateral GCT and five had metachronous bilateral GCT. DNA extracted from these FFPE blocks was analyzed using WES and RRBS. Out of approximately 20,000 genes investigated, 189 (<1%) had a detectable mutation in the nine paired cases. This is similar to other studies that found a low number of somatic mutations in GCT [7,8]. We found that two patients with synchronous disease had a similar mutation in both the right and left tissue (SP2, SP7); however, the other two patients with synchronous disease did not (SP4 and SP5). For the five patients with metachronous disease, we did not find a similar mutation in the right and left GCT tissues. For most of these patients, chemotherapy or radiation therapy did not occur prior to resection. However, for one patient SP6, chemotherapy was provided after the right GCT (SP6R) was removed, but this was approximately 66.2 months prior to SP6L diagnosis. Although the mutations caused by chemo- or radiation therapy could possibly cause GCT in the contralateral testis, it was not apparent in this analysis or in our previous analysis of patients with subsequent malignant neoplasms with a prior history of GCT that this occurred [25].

We applied RRBS [19,20] to map methylated cytosines in the left and right GCT tissues. Using a hierarchical cluster dendrogram, we found that nonseminomas and seminomas clustered separately. In our study, bilateral nonseminomas SP3 and SP6 were metachronous but still cluster together. In contrast, two synchronous bilateral seminomas (SP2 and SP4) did not cluster, although another (SP7) did.

Due to the separation in space and time, bilateral GCT provides us with a unique opportunity to trace and track its distinct genetic and/or epigenetic origins owing to the well-known migratory pathways and developmental timelines of its putative cancer-initiating primordial germ cell (PGC). A prevailing theory envisions the clonal evolution of germ cell tumors from a seminoma to a nonseminoma [26]. Another theory suggests that a malignant precursor develops into either a seminoma or a nonseminoma [27]. We proposed an alternative viewpoint in which discrete precursor cells in a stem-cell hierarchy give rise to either a nonseminoma (that may contain a seminoma) or a seminoma [28]. 

We postulated that nonseminomas tend to express a more heterogeneous and widely metastatic phenotype because they originate from earlier, more pluripotent gonadal stem cells. In contrast, pure seminomas display a more homogeneous and limited metastatic phenotype (i.e., being limited to the lymph nodes) since they are derived from later gonadal stem cells. The observation that nonseminomas (especially those that comprise mixed yolk sac tumor and seminoma components) show a greater tendency to undergo somatic transformation, while pure seminomas do not, is also consistent with these inherent biological inclinations and clinical presentations [29].

This model of tumorigenesis is compatible with an age-dependent spectrum of GCT [30] in which pediatric germ cell tumors (Type I) tend to be teratoma and yolk sac tumors, alluding to their origin in an earlier stem cell or PGC. Seminoma (Type IIa) likely engenders from later PGCs and occurs in older patients (30 to 40 years old) compared to nonseminomas (Type IIb), which we surmise are derived from earlier PGCs and are more commonly seen in younger patients (20 years old). In contrast, Type III GCT, comprising spermatocytic seminoma, arises from more terminal differentiated primordial germ cells before the formation of spermatozoa in elderly patients (>50 years old).

When a mixed GCT displays concordance in the genetic signature of its distinct pathological components (e.g., embryonal carcinoma vs. teratoma), there is evidence of a common clonal origin [31,32,33]. However, it is not apparent whether this clonal origin is caused by a cellular (stem cell) aberrations or genetic defects. In addition, Stevens demonstrated that different germ cells in the stem cell hierarchy have different malignant potentials [34,35]. He suggested a stem cell rather than genetic origin of cancers in GCT by demonstrating that mice lacking primordial germ cells failed to develop teratoma despite carrying the putative germline genetic mutations (i.e., Sl^j^/Sl^j^ and Sl^d^/Sl^d^) that should otherwise have led to the formation of teratoma [34]. 

The right and left specimens of the nonseminomas SP3 and SP6 presented similar DNA methylation profiles, as seen by their close proximity in the unsupervised hierarchical clustering analysis, which suggests that their PGC of origin likely precedes the separation and migration of germ cells to the right and left gonadal ridges. Furthermore, our data suggest that nonseminomas may harbor fewer repeated mutations (i.e., five samples containing one mutation (ATM)). These results are consistent with the idea that the cellular context trumps the genetic content; when a cancer-initiating cell has more stemness properties earlier in the stem-cell hierarchy (i.e., greater pluripotency), it has greater potential to become malignant (including greater heterogeneity), and requires fewer driver genetic defects or aberrations to be so [36,37].

In contrast, the right and left specimens of the two synchronous bilateral seminomas (SP2 and SP4) do not cluster together, suggesting that their PGC formed after the separation into the right and left gonadal ridges occurred. Most seminomas had a higher mutational burden than the nonseminomas, which was not dependent on the synchronous or metachronous nature of the disease. We generated a model for the PGC for each patient based on their DNA methylation pattern and genomic mutations (Figure 3). We suspect that mixed nonseminoma with higher proportion of embryonal carcinoma tends to originate from a PGC higher up in the hierarchy, while pure seminoma tends to arise from a PGC that is lower in the hierarchy [38]. A dissimilar epigenomic profile (higher clustering distance) and genomic signature (discordance) would suggest a separate clonal origin in the formation of the SP57R, SP5R, SP2R, SP4R, and SP74R in the right testis and SP57L, SP5L, SP2L, SP4L, and SP74L in the left testis, after PGCs separate and migrate into the right and left gonadal ridges, respectively.

This study provides further insights into the origin of other bilateral malignancies in addition to GCT. For instance, a preponderance of synchronous bilateral ovarian carcinomas displayed a similar number of genomic abnormalities and mutational signatures [39,40,41]. However, when occasional discordances in the genomic abnormalities and mutational signatures did occur between synchronous bilateral ovarian carcinomas, this was attributed to independent cellular origins rather than a monoclonal origin [42,43]. 

A stem cell rather than genetic origin of cancer would explain why women for whom cancer was found in the second breast within five years of a first breast cancer diagnosis were more likely to die, whereas women diagnosed with contralateral breast cancer more than 10 years after their first diagnosis were not [44]. An independent cellular origin could also solve the paradox of why breast cancer survivors who develop a second breast cancer in the contralateral breast have a higher risk of death, and preventing that cancer with prophylactic surgery does not change the outcomes [45].

We found significantly higher DNA methylation in the nonseminoma compared to seminoma tissues at least in one site of nine genes. Higher CpG methylation was observed in nonseminoma cell lines compared to a seminoma cell line [46] and nonseminoma versus seminoma pediatric GCT [47]. For the gene *TPPP1*, higher methylation in nonseminomas was found at two different sites. TPPP was previously shown to be associated with testicular cancer. TPPP is a protein family with three members in mammalian cells: TPPP1, TPPP2, and TPPP3. Interestingly, TPPP1 was found to be brain-specific while TPPP2 is highly expressed in the testes [48]. TPPP2 was found via proteomic analysis to be involved in testicular cancer [49]. The other eight genes found to have higher methylation in nonseminomas versus seminomas have not been specifically studied in testicular cancer, but their protein expression has been shown to be associated with general carcinogenesis. ARC is a potent inhibitor of apoptosis that was linked to stemness by promoting cell survival and proliferation [50]. High ARC levels can help maintain a stem-like cell population with an increased self-renewal potential. JRK is a positive regulator of β-catenin transcriptional activity and is overexpressed in colon, breast, and ovarian cancer [51]. The depletion of JRK represses β-catenin transcriptional activity and reduces cell proliferation. MATK is a protein that plays a role in regulating stemness and was found to be hypermethylated in colorectal cancer, which lowers its expression [52]. MATK can inhibit SRC kinase activity and the loss of MATK is hypothesized to promote cancer progression in colorectal cancer. ADARB2 is also thought to effect stemness and was found to be a possible tumor-suppressor in papillary thyroid cancer [53] and brain cancers [54]. Reduced RNA expression of ADARB2 was found in low-grade astrocytomas, anaplastic astrocytomas, glioblastomas, and oligodendrogliomas compared to normal brain tissue, with the lowest expression found in glioblastomas [54]. RAP1GAP2 can indirectly influence stem cell behavior, particularly in the context of maintaining the stem cell niche and regulating differentiation pathways in certain cell types. CRHR2, INMT, and GALNT9 expression have been shown to be associated with cancer and may play a role in regulating the self-renewal and proliferation of cancer stem cells [55,56,57,58]. Interestingly, INMT expression was found to be lower in early-stage prostate cancer [57] but increased in castration-resistant prostate cancer [56]. Our differential methylation data suggest potential targets in nonseminoma testicular cancer that can be further explored for biomarker development, prognosis, or therapy.

In addition, we specifically looked at the methylation differences in *AR*, *JUP*, and *MAGE-A4,* as these genes were found in other studies to be associated with testicular cancer [22,23,24,59,60,61]. We found differential methylation between the 11 seminomas and 7 nonseminomas in *MAGE-A4*. Higher hypermethylation was found in five separate sites in *MAGE-A4,* and less prominently in one site for *JUP,* in the nonseminoma tissues. This correlates with the immunohistochemistry results of Aubry et al. [23], who found MAGE-A4, a germ cell-specific marker, to be uniformly expressed in classical seminomas and not in anaplastic seminomas or nonseminomas. Other tissue microarrays demonstrated two molecular types of seminomas that can be distinguished using markers such as CD30, AE 1/3 (anion exchanger 1/3), and JUP [62].

These significant differences in DNA methylation between nonseminomas and seminomas in our data could be due to the bilateral nature of our samples and might not be found in non-bilateral testicular cases. In addition, seminomas tend to be homogeneous morphologically, whereas 80% of nonseminomas are mixed tumors that can include embryonal, YST, teratoma, and seminoma regions. Further work would have to be performed to determine if the same methylation differences are observed in non-bilateral testicular cases and within mixed as well as non-mixed nonseminonas. In addition, this study is constrained by the rarity of bilateral GCT. Tumor samples are often no longer available at different facilities after prolonged storage periods, making obtaining patient consent for research purposes a difficult endeavor.

Whether genetic mutation and/or cellular context are causative is central in our fundamental understanding of the origin and nature of cancers and cannot be solved using the few but representative cases in this exploratory study, which, to our knowledge, is the first of its kind. Nevertheless, these results suggest that genomic data may supplement, if not complement, epigenomic findings to elucidate a stem-cell versus genetic origin and the nature of GCT and cancers in general. Additional cases need to be examined, and it is our hope that other institutions will perform similar studies to validate our findings, as this area of research holds the key to transforming our understanding of cancer biology and approach to treatment.

## 5. Conclusions

In this single-institution study of 38 patients with a bilateral germ cell tumor of the testis, no significant difference in overall survival was observed between those with metachronous and synchronous disease, and no significant difference was observed in any identifiable categorical variables. WES and RRBS were performed on paired left and right GCT tissues from nine patients. Genomic analysis revealed no notable differences in genomic mutation profiles between synchronous and metachronous tumors. Methylation profiling showed a distinct clustering according to histology into seminoma and nonseminoma, with nonseminomas exhibiting higher levels of CpG methylation. Overall, these findings suggest that the cellular context, rather than specific genomic alterations, may play a more critical role in the pathogenesis of bilateral testicular cancer.

## Figures and Tables

**Figure 1 cells-14-00658-f001:**
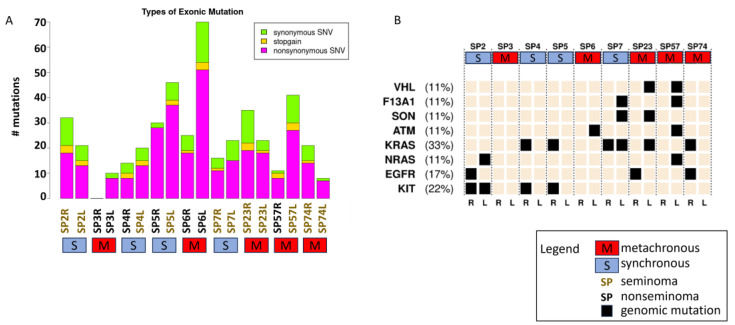
WES data (**A**) Single-nucleotide variants (SNV) found in the genes within each sample. Black text represents nonseminoma and brown text represents seminoma. (**B**) Top seven genes mutated in nine patients’ bilateral right (R) and left (L) GCT tissues. Synchronous GCT is represented by a blue box labeled with S, while metachronous GCT is represented by a red box labeled with M. Genomic mutations are represented by a black box.

**Figure 2 cells-14-00658-f002:**
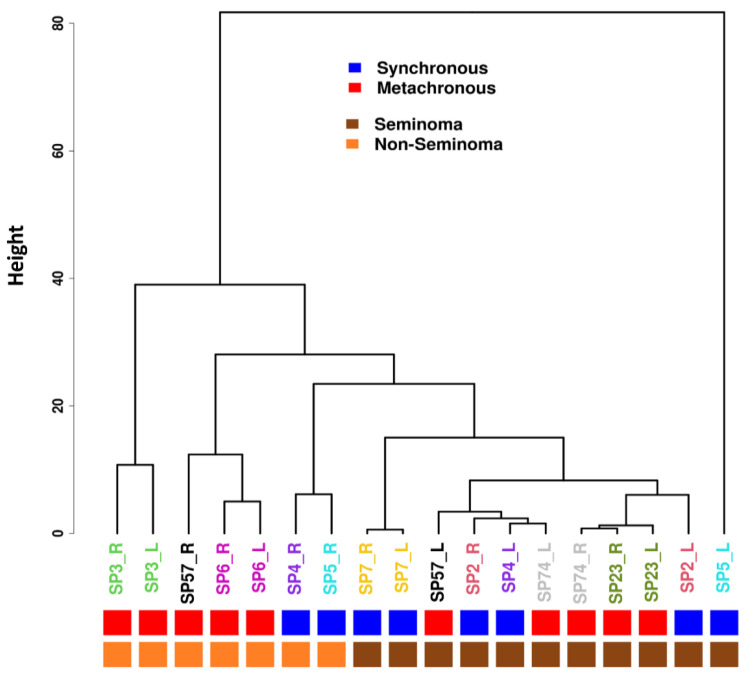
Hierarchical cluster dendrogram highlighting the clustering of methylation profiles in the tumor samples.

**Figure 3 cells-14-00658-f003:**
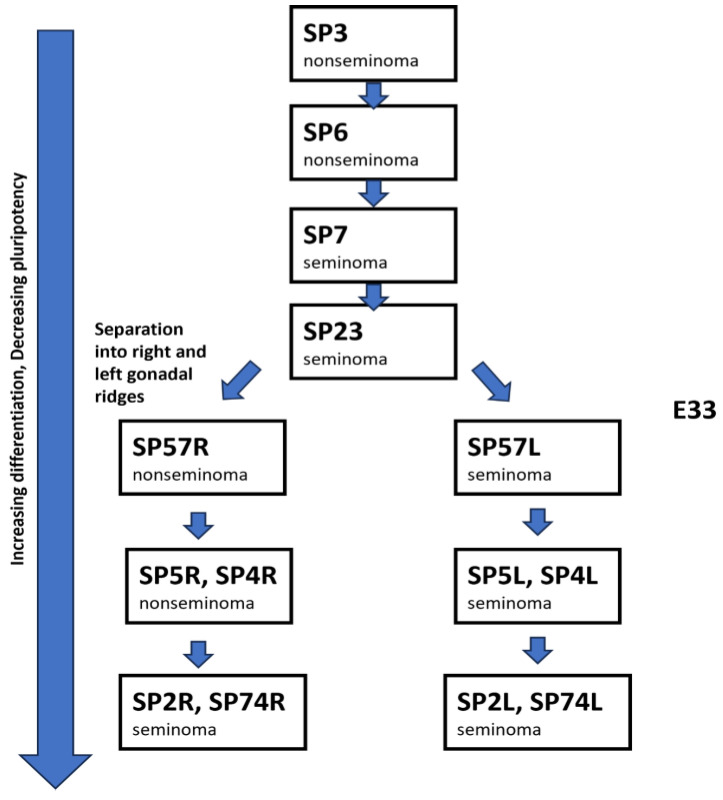
Model of where in the PGC hierarchy each patient’s cancer-initiating stem cell occurred. PGCs further down the hierarchy have less pluripotency and increased differentiation compared PGCs further up the hierarchy. We suspect a clonal origin of SP57R, SP5R, SP2R, SP4R, and SP74R in the right testis and SP57L, SP5L, SP2L, SP4L, and SP74L in the left testis, which occurs after PGCs separate and migrate into the right and left gonadal ridges. E33 corresponds to human embryonic day 33.

**Table 1 cells-14-00658-t001:** Descriptive statistics of patient population (n (%N)) with Fisher’s exact test *p*-value provided to compare categorical variables, and Mann–Whitney U test *p*-value provided to compare continuous variables between synchronous and metachronous disease.

	Total (n = 38)	Synchronous (n = 7)	Metachronous (n = 31)	*p*-Value
Race/Ethnicity (%)				
Caucasian/White	27 (71.1)	5 (71.4)	22 (71.0)	>0.99
African American/Black	10 (26.3)	2 (28.6)	8 (25.8)	
Hispanic	1 (2.6)	0 (0)	1 (3.2)	
Median Follow-Up (Months; IQR)	134.2 (69.5–222.1)	50.3 (22.8–99.7)	157.3 (91.8–252.0)	0.004
Median Age at 1st Orchiectomy (IQR)	27. 5 (20–32)	26 (20–34)	27.5 (20–31)	0.77
Median Age at 2nd Orchiectomy (IQR)	33 (26–37)	26 (20–34)	33 (29–37)	0.11
Stage	I—28 (73.7)	I—6 (85.7)	I—22 (71.0)	0.65
	II—9 (23.7)	II—1 (14.3)	II—8 (25.8)	
	III—1 (2.6)		III—1 (3.2)	
Pathologic Subtype (%)				
Bilateral Seminoma	13 (34.2)	3 (42.9)	10 (32.3)	0.66
Bilateral Nonseminoma	12 (31.6)	1 (14.3)	11 (35.5)	
Seminoma/Nonseminoma	13 (34.2)	3 (42.9)	10 (32.3)	
Disease Progression (%)				
Yes	7 (18.4)	0 (0)	7 (22.6)	0.31
No	31 (81.6)	7 (100)	24 (77.4)	
Survival (%)				
Alive	36 (94.7)	7 (100)	29 (93.5)	>0.99
Dead	2 (5.3)	0 (0)	2 (6.5)	

**Table 2 cells-14-00658-t002:** Tumor histopathologic characteristics, interval between tumors, and treatment.

Sample ID	Laterality	Size (cm)	Interval (Months)	Stage	Treatment	Pathology
SP2_L_S2	Left	7.8	S—0	I	XRT *	Seminoma
SP2_R_S1	Right	5.6	S—0	I		Seminoma
SP3_L_S4	Left	1.2	M—21.2	I	BEP (X2)	Nonseminoma (20% embryonal carcinoma, 20% yolk sac tumor, 60% teratoma)
SP3_R_S3	Right	N/a	M—0	I		Nonseminoma (20% yolk sac tumor, 80% teratoma)
SP4_L_S6	Left	2.6	S—0	I	BEP (x3)	Seminoma
SP4_R_S5	Right	5	S—0	I		Nonseminoma (10% embryonal carcinoma, 90% seminoma)
SP5_L_S8	Left	3.5	S—0	I	BEP (x3)	Seminoma
SP5_R_S7	Right	9.2	S—0	I		Nonseminoma (85% yolk sac tumor, 10% immature teratoma, 5% embryonal carcinoma)
SP6_L_S10	Left	2.7	M—66.2	I	VIP (x3)	Nonseminoma (90% embryonal carcinoma, 5% seminoma, 4% teratoma, 1% yolk sac tumor)
SP6_R_S9	Right	2.8	M—0	II	BEP (x3)	Nonseminoma (90% embryonal carcinoma, 10% yolk sac tumor)
SP7_L_S12	Left	2.9	S—0	I	None	Seminoma
SP7_R_S11	Right	2.2	S—0	I		Seminoma
SP23_L_S14	Left	2.7	M—0	I	XRT	Seminoma
SP23_R_S13	Right	3.0	M—10	I		Seminoma
SP57_L_S16	Left	2.9	M—158.3	I	None	Seminoma
SP57_R_S15	Right	3.1	M—0	I		Nonseminoma (80% embryonal carcinoma, 10% teratoma, 5% choriocarcinoma, 4% seminoma, 1% yolk sac tumor)
SP74_L_S18	Left	1.5	M—18.8	I	XRT	Seminoma
SP74_R_S17	Right	0.9	M—0	I		Seminoma

* XRT was suggested for treatment but could not be verified by external institution.

**Table 3 cells-14-00658-t003:** Highest methylation sites found between nonseminoma and seminoma tissues, with a >40% mean difference in methylation. The chromosome, the position of the methylation, the mean methylation value for nonseminomas, the mean methylation value for seminomas, the mean difference in methylation, the adjusted *p*-value, and the gene are presented in the table. LOC represents uncharacterized RNA genes.

Chromosome	Position	Mean Nonseminoms	Mean Seminomas	Mean Difference	*p*-Value	Gene
8	144000000	79.46	34.56	44.90	<0.0001	*ARC*, *JRK*
19	3778280	67.81	20.41	47.40	<0.0001	*MATK*
10	1701540	73.76	28.79	44.97	<0.0001	*ADARB2*
5	677993	80.97	35.25	45.72	0.0001	*TPPP1*
5	678007	87.82	42.96	44.87	0.0003	*TPPP1*
7	56620283	74.92	28.81	46.12	0.0003	*LOC101928401*, *LOC401357*
7	30765560	76.14	28.98	47.16	0.0003	*CRHR2*, *INMT*
17	2699778	52.22	6.27	45.95	0.0004	*RAP1GAP2*
12	133000000	83.65	37.61	46.04	0.0004	*GALNT9*
17	7569655	76.20	29.12	47.08	0.0005	*LOC100507351*, *LOC100132174*

**Table 4 cells-14-00658-t004:** Significant differential methylation between nonseminomas and seminomas in *MAGE-A4* and *JUP*. The chromosome, the position of the methylation, the mean methylation value for nonseminomas, the mean methylation value for seminomas, the mean difference in methylation, the adjusted *p*-value, and the gene are shown in the table.

Chromosome	Position	Mean Nonseminomas	Mean Seminomas	Mean Difference	*p*-Value	Genes
X	151081290	82.31	35.96	46.34	0.001	*MAGE-A4*
X	151081284	81.97	34.80	47.17	0.003	*MAGE-A4*
X	151081291	85.49	35.97	49.52	0.008	*MAGE-A4*
X	151081297	1.74	0	1.74	0.018	*MAGE-A4*
X	151081296	82.31	35.96	47.95	0.031	*MAGE-A4*
17	39942776	81.97	34.80	35.47	0.013	*JUP*

## Data Availability

The data supporting the findings of this study are available upon reasonable request from the corresponding authors.

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
