# Peer review of "Bilateral Germ Cell Tumor of the Testis: Biological and Clinical Implications for a Stem Versus Genetic Origin of Cancers"

_cells, 2025, doi:10.3390/cells14090658_

Round 1
Reviewer 1 Report
Comments and Suggestions for Authors
This article mainly focuses on exploring the origin of tumors according to the testicular tumor, which is a very interesting topic and the article is very novel. However, there are some questions that need to be explored.
- The retrieval time for the sample collection in this article is up to 2/28/2021. So far, 4 years have passed. Can more new cases be added?
- This article lacks a conclusion.
- Among the 38 patients collected, there were only 7 simultaneous patients and 31 metachronous patients. The difference is large and not very matched. Could you consider balancing the number of patients in the two groups?
- Could you add patient tumor staging information? Tumors in different stages have different prognoses.
- Table 1 shows that the disease progression in the two groups is 0% and 22.6%, respectively, which is very different, but further analysis is missing.
- Table 1 shows that two people died in the metachronous group, but further analysis is missing.
- In the survival analysis in the supplementary materials, there is no method for it and the analysis results like HR value also lack.
- This article mentioned that spermatogonia and non-spermatogonia have obvious methylation, but different groups have adopted some chemotherapy methods. Chemotherapy may affect methylation. How to avoid it?
English is good especially taking intoi account that this is an open access journal which can a little bit improve English.
Author Response
We want the thank the reviewers for their thorough review of the manuscript. Their suggestions have significantly improved the manuscript. Our answers to reviewers' questions are in red font below and, we have highlighted the added text in the manuscript based on their suggestions in yellow.
Reviewer 1
This article mainly focuses on exploring the origin of tumors according to the testicular tumor, which is a very interesting topic and the article is very novel. However, there are some questions that need to be explored.
- The retrieval time for the sample collection in this article is up to 2/28/2021. So far, 4 years have passed. Can more new cases be added?
To have sufficient time for follow-up (4-5 years), we have only used samples from patients that received treatment before or on 2/28/2021.
- This article lacks a conclusion.
We have added a conclusion to the manuscript.
- Among the 38 patients collected, there were only 7 simultaneous patients and 31 metachronous patients. The difference is large and not very matched. Could you consider balancing the number of patients in the two groups?
Testicular cancer is a rare (less than 1% of all cancers diagnosed in men), and bilateral testicular cancer is even more rare (less than 2% of testicular GCT tumors). This rarity makes it unfeasible to balance the number of metachronous versus synchronous patients in our study. In addition, the patient is specified to have a synchronous bilateral GCT based on if the bilateral orchiectomies are performed on the same date. We would like to emphasize that this is common parlance in clinical practice and may not reflect biological significance (lines 243-244), because some patients will have their metachronous bilateral GCT diagnosed and removed synchronously. This is in part due to GCT tumor growth rate being highly variable. For example, seminoma, teratoma, and yolk sac tumors tend to grow slowly and may be asymptomatic until they become palpable. In contrast, embryonal or choriocarcinoma tend to grow quickly, with an estimated doubling time of 10 to 30 days (PMID 1694087).
Could you add patient tumor staging information? Tumors in different stages have different prognoses.
We have added tumor staging to Table 1 and Table 2.
- Table 1 shows that the disease progression in the two groups is 0% and 22.6%, respectively, which is very different, but further analysis is missing.
We have now run Fisher’s exact test to determine the p-value between synchronous and metachronous disease in our 38-patient cohort. No statistical significance is observed in disease progression between the two groups.
- Table 1 shows that two people died in the metachronous group, but further analysis is missing.
Hazard ratio and median survival time were not calculated between the two groups as one group had zero events.
- In the survival analysis in the supplementary materials, there is no method for it and the analysis results like HR value also lack.
We have now added the paragraph below to the methods section.
Added to Text. For categorical variables, Fisher’s exact test was used to determine the p-value between synchronous and metachronous disease. For continuous variables, Mann Whitney U test (also called Mann-Whitney Wilcoxon Test) was used to determine the p-value between synchronous and metachronous disease. The log-rank p-value was determined between the overall survival Kaplan-Meir curves of synchronous and metachronous disease in our 38-patient cohort. Hazard ratio and median survival times between synchronous and metachronous disease was not calculated as one group had zero events.
- This article mentioned that spermatogonia and non-spermatogonia have obvious methylation, but different groups have adopted some chemotherapy methods. Chemotherapy may affect methylation. How to avoid it?
Almost always, orchiectomy is performed before chemotherapy or radiation therapy. In those rare cases of high volume symptomatic GCT cases when chemotherapy needs to be given before orchiectomy, either no viable tumor or teratoma remain, so they are no longer relevant and were not included for the objectives of this study.
Added to text: Orchiectomies are often performed before chemotherapy or radiation therapy. Patients are given chemotherapy or radiation therapy prior to resection only in rare cases of high volume symptomatic GCT. These patients were excluded from our study. Extensive staging is performed prior to any orchiectomy, which includes ultrasound of the contralateral testis, therefore confirming that in metachronous disease the other GCT lesion was not present by imaging at the time of the initial orchiectomy.
For most of these patients, chemotherapy or radiation therapy did not occur prior to resection. However, for one patient SP6, chemotherapy was given after the right GCT (SP6R) was removed but this was approximately 66.2 months prior to SP6L was diagnosed.
Reviewer 2 Report
Comments and Suggestions for Authors
Whether tumor phenotypes are driven by cellular context (e.g., stem-ness origin) or genetic content (e.g., genomic aberrations) is a debated topic that adds to the heterogeneity we see in tumors. The authors set out to use a unique system, germ cell tumors of the testis (GCTs), to study this dichotomy. Their samples were histologically either seminomas or nonseminomas, and temporally, they were either synchronous (found simultaneously) or metachronous (found at different times). The space and time separation allowed the authors to link back to the developmental origin as coming likely from a cellular or genetic context. This data is not easily accessible, so it does provide a unique system that only these authors were able to use, adding to their importance.
The authors used whole exome sequencing (WES) to test the genetic-origins-hypothesis and reduced representation bisulfite-sequencing (RBBS) to test the cellular-origins-hypothesis. The authors found that – in line with other studies – they had few mutations across the samples. Still, there was a mix of mutations that were/were not shared across the bilateral sides (Fig.1). The authors used the RBBS to hierarchically cluster the samples by epigenetic similarity, arguing that when the samples were clustered near each other the cell of origin hypothesis was likely driving the development of the tumor (Fig. 2). Together, the dissimilar epigenomic profile (higher clustering distance) and genomic signature (discordance) would suggest a separate clonal origin in some of the tumors. Lastly, the authors found several differentially methylated genes between seminoma and non-seminoma bilateral GCTs, including a few related to cancer pathways and germ cell-specific development (Table 4&5).
Overall, this paper provides context for a rare phenotype of GCTs, bilateral GCTs of the testis. This unique situation also provided the authors an opportunity to explore the genetic vs cellular diversity of each of the contralateral tumor development. I think that there are a few places where we can clarify the logic that led to the conclusion, especially surrounding Fig. 4 – the diagram of the hypothesized developmental timepoint of tumor origination. I provide these suggestions below:
Major Points:
- Are there any limitations to FFPE whole genome sequencing? You cite the work you have done with citation #12, however we have seen in the past evidence of micro inversions. Here would these limit your ability to call mutations across samples? Especially in 2 organs?
- Several of your samples (SP3, SP6, and SP74) all had treatment (BEP, VIP/BEP, and XRT – respectively). Is there any evidence of XRT or BEP after the timing of treatment in the WGS? And are you at all worried about these affecting mutation calling?
- Table 1: In line 267, you write, “… There were 13 bilateral seminomas, 14 bilateral nonseminomas, and 11 bilateral seminomas and nonseminoma”. However, in Table 3, you have 13 seminoma, 12 nonseminoma, and 13 seminoma/nonseminoma. Please clarify.
- Table 4: All the differential methylated mean differences are negative. From my understanding of your data, this means that it is higher in nonseminomas. However, are there any genes that were positive mean difference? i.e. Higher in seminomas?
- Line 255: You looked at AR, MAGE-A4, and JUP. However, you did not bring it up in Table 5, is it just not differentially methylated?
- Line 321: You wrote: “as seen by their close proximity in the unsupervised hierarchical clustering analysis, which suggests that their PGC of origin likely precedes the separation and migration of germ cells to the right and left gonadal ridges.” Can you please explain why you would not expect these methylation patterns to change over differentiation?
Minor Points:
- In your methods you write Reduce Representation Bisulfite-Seq as RRBS, but in the paper you use RBBS. Is there a reason for RBBS vs. RRBS (RRBS being the way I might expect it as a reader?)
- In line 45, you use the word “prototype” to describe the germ cell as a stem cell. What do you mean here and why do you use the word prototype?
- Line 163-164: You wrote no difference among the survival of synchronous and metachronous GCTs. You also write the same thing in line 269. But I did not see statistics and p-value, could you please provide that?
- Line 166: You write that the overall survival is similar to other patients, with citation 18, what is the expected survival in that data?
- Figure 1A: The figure is difficult to follow with the current use of colors (black text-seminoma, brown text-nonseminoma), I suggest adding a small legend below the y-axis. Similarly, the color of synchronous vs metachronous would do well with a legend.
- Line 203: In Supp. Figure 2, SP3 falls off the line of the mapping efficiency. Do you expect this to change your analyses at all or your interpretation?
- Line 207-210: “We found that the pairs of metachronous nonseminomas (i.e. patients SP3 and SP6) displayed a similar methylation profile but did not share any genetic mutations, suggesting a common cellular rather than genetic origin despite a separation in time of their diagnoses by 21 and 67 months, respectively”
- I agree that metachronous and seminoma do not share genetic origin, but I just need a justification written about how/why you think this as a reader.
- Line 233 & 237: You used adjusted p-value < 0.01 for the seminoma vs. nonseminoma and nominal p-value <0.001 for left vs. right. Why did you not just use adjusted p-value or nominal?
Overall, I believe the authors did a well-designed experiment. I believe that the authors will provide an important and interesting contribution to the field after responses to the comments above. The authors’ justification for using GCTs as their model to answer their question about cellular vs. genetics origins is spot on and I commend them on their approach. I also found their summary of experiments in the first few paragraphs of the Discussion section to be superb in capturing the important context of the results. I may gently urge the authors to use some of the writing from the Discussion section to contextualize the Results section paragraphs.
Author Response
We want the thank the reviewers for their thorough review of the manuscript. Their suggestions have significantly improved the manuscript. Our answers to reviewers' questions are in red font below and, we have highlighted the added text in the manuscript based on their suggestions in yellow.
Reviewer 2.
Whether tumor phenotypes are driven by cellular context (e.g., stem-ness origin) or genetic content (e.g., genomic aberrations) is a debated topic that adds to the heterogeneity we see in tumors. The authors set out to use a unique system, germ cell tumors of the testis (GCTs), to study this dichotomy. Their samples were histologically either seminomas or nonseminomas, and temporally, they were either synchronous (found simultaneously) or metachronous (found at different times). The space and time separation allowed the authors to link back to the developmental origin as coming likely from a cellular or genetic context. This data is not easily accessible, so it does provide a unique system that only these authors were able to use, adding to their importance.
The authors used whole exome sequencing (WES) to test the genetic-origins-hypothesis and reduced representation bisulfite-sequencing (RRBS) to test the cellular-origins-hypothesis. The authors found that – in line with other studies – they had few mutations across the samples. Still, there was a mix of mutations that were/were not shared across the bilateral sides (Fig.1). The authors used the RBBS to hierarchically cluster the samples by epigenetic similarity, arguing that when the samples were clustered near each other the cell of origin hypothesis was likely driving the development of the tumor (Fig. 2). Together, the dissimilar epigenomic profile (higher clustering distance) and genomic signature (discordance) would suggest a separate clonal origin in some of the tumors. Lastly, the authors found several differentially methylated genes between seminoma and non-seminoma bilateral GCTs, including a few related to cancer pathways and germ cell-specific development (Table 4&5).
Overall, this paper provides context for a rare phenotype of GCTs, bilateral GCTs of the testis. This unique situation also provided the authors an opportunity to explore the genetic vs cellular diversity of each of the contralateral tumor development. I think that there are a few places where we can clarify the logic that led to the conclusion, especially surrounding Fig. 4 – the diagram of the hypothesized developmental timepoint of tumor origination. I provide these suggestions below:
Major Points:
- Are there any limitations to FFPE whole genome sequencing? You cite the work you have done with citation #12, however we have seen in the past evidence of micro inversions. Here would these limit your ability to call mutations across samples? Especially in 2 organs?
Multiple studies have demonstrated high concordance in mutation detection between formalin-fixed paraffin-embedded (FFPE) and fresh-frozen tissues using whole-exome sequencing (WES). Examples are: (PMID: 33435284) a 2021 analysis of 16 matched gastro-oesophageal tumor pairs found 97% median mutational concordance (range 80.1–98.68%) when applying thresholds of ≥10 mutant reads and ≥5% variant allele frequency; (PMID: 26641479) a 2015 WES study of four cancer patients showed comparable somatic mutation calls between FFPE and frozen pairs despite FFPE DNA fragmentation; and (PMID: 26305677) a separate 2015 analysis of 13 Flash frozen/FFPE pairs demonstrated >98.9% base call concordance in WES data, with FFPE samples achieving similar SNV detection accuracy to frozen samples when quality filters were applied. In addition, the concern regarding FFPE primarily relates to the artifacts it may introduce. The fact that the low levels of mutations and its relevant genes align well with expectation helps to mitigate this concern.
- Several of your samples (SP3, SP6, and SP74) all had treatment (BEP, VIP/BEP, and XRT – respectively). Is there any evidence of XRT or BEP after the timing of treatment in the WGS? And are you at all worried about these affecting mutation calling?
Orchiectomy is normally performed before chemotherapy (SP3L and SP6R) or radiation therapy (SP74L), and this is an important inclusion criterion for the purposes of this study. It is true that chemotherapy (BEP x3) was given after SP6R but 66.2 months before SP6L was diagnosed. Since extensive staging (including ultrasound of the left testis) is performed at the time of diagnosis, SP6L was not present 66.2 months before the initial diagnosis of GCT in the right testis. Although mutations caused by chemotherapy and radiation may cause GCT in the contralateral testis, it is not apparent from our analysis in this limited study, and from another study of subsequent malignant neoplasms of patients with a prior history of GCT who have received chemotherapy and/or radiation therapy (PMID: 33327406).
Added to text: For most of these patients, chemotherapy or radiation therapy did not occur prior to resection. However, for one patient SP6, chemotherapy was given after the right GCT (SP6R) was removed but this was approximately 66.2 months prior to SP6L was diagnosed. Although mutations caused by chemo- or radiation therapy could possibly cause GCT in the contralateral testis, it is not apparent in this analysis or in our previous analysis of patients with subsequent malignant neoplasms with a prior history of GCT that this occurs (PMID: 33327406).
- Table 1: In line 267, you write, “… There were 13 bilateral seminomas, 14 bilateral nonseminomas, and 11 bilateral seminomas and nonseminoma”. However, in Table 3, you have 13 seminoma, 12 nonseminoma, and 13 seminoma/nonseminoma. Please clarify.
Thank you for detecting these typos. We have corrected the error.
Table 4: All the differential methylated mean differences are negative. From my understanding of your data, this means that it is higher in nonseminomas. However, are there any genes that were positive mean difference? i.e. Higher in seminomas?
Yes, you are correct the CpG methylation is higher in nonseminomas. We have now clarified this in the text and added a reference that observes a similar difference hypermethylation in nonseminoma cell lines (PMID: 29263807) and pediatric GCT (PMID: 23806198).
Added to text: We adopted a cutoff of > 40% mean difference in methylation between groups with adjusted p-values < 0.01 to identify differential methylation. We found 894 CpG sites hypermethylated in nonseminomas and none in seminoma tissues based on our criteria. The top 10 based on p-values are shown in Table 3. Based on an adjusted p-value of 0.01 with >10% mean difference in methylation, the highest hypermethylation in seminomas versus nonseminomas was found at one CpG site: chromosome 7, position 148787751, gene ZNF786, with a mean difference of 10.8. A similar analysis was performed to determine the differential methylation between right and left GCT in the 9 pairs with > 40% mean difference in methylation with adjusted p-value of < 0.01 and no genes were found.
We found significantly higher DNA methylation in the nonseminoma compared to seminoma tissues at least in one site of nine genes. Higher CpG methylation has been observed in nonseminoma cell lines compared to a seminoma cell line (PMID: 29263807) and nonseminoma versus seminoma pediatric GCT (PMID: 23806198).
- Line 255: You looked at AR, MAGE-A4, and JUP. However, you did not bring it up in Table 5, is it just not differentially methylated?
No significantly differentiated CpG sites were detected in AR and therefore it is not included in the table.
Added to text: AR methylation on CpG sites was not statistically significant between seminoma and nonseminoma tissues.
- Line 321: You wrote: “as seen by their close proximity in the unsupervised hierarchical clustering analysis, which suggests that their PGC of origin likely precedes the separation and migration of germ cells to the right and left gonadal ridges.” Can you please explain why you would not expect these methylation patterns to change over differentiation?
Because differentiation and heterogeneity (to teratoma in a nonseminoma) are already evident in SP3 and SP6 (see Table 2) and which is not the case for SP2 and SP74 (pure seminomas), our data is more consistent with the hypothesis that separate clones in the gonadal germ cell hierarchy form nonseminomas vs seminomas (references 26 and 27) rather than the alternative hypothesis in which seminomas differentiate into nonseminomas (reference 25).
Minor Points:
- In your methods you write Reduce Representation Bisulfite-Seq as RRBS, but in the paper you use RBBS. Is there a reason for RBBS vs. RRBS (RRBS being the way I might expect it as a reader?)
We have now consistently used RRBS in the text.
- In line 45, you use the word “prototype” to describe the germ cell as a stem cell. What do you mean here and why do you use the word prototype?
Since germ cells are the origin of all fetal cells, germ cell tumors serve as the prototype or archetype of stem cell tumors, providing valuable insights in the role of differentiation in cancer development. We discuss this more in depth in our publication in Cancers (PMID: 33562202).
Added to text: Since germ cells are the origin of all fetal cells, a germ cell is a prototype stem cell and GCT is the archetype of stem cell tumors with relatively few but putatively pivotal genetic defects (e.g., i[12p]).
- Line 163-164: You wrote no difference among the survival of synchronous and metachronous GCTs. You also write the same thing in line 269. But I did not see statistics and p-value, could you please provide that?
We have now added a p-value to the Kaplan Meier curve.
- Line 166: You write that the overall survival is similar to other patients, with citation 18, what is the expected survival in that data?
We have now added new text and cited a more recent study that determined the outcomes of 1111 patients with either synchronous or metachronous GCT.
Added to text. Our results are similar to a recent 1111 patient cohort with either synchronous or metachronous GCT (18), which found no differences in baseline clinical characteristics or statistical difference in overall survival between groups (log-rank p-value of 0.62).
- Figure 1A: The figure is difficult to follow with the current use of colors (black text-seminoma, brown text-nonseminoma), I suggest adding a small legend below the y-axis.
We have now added a legend.
- Line 203: In Supp. Figure 2, SP3 falls off the line of the mapping efficiency. Do you expect this to change your analyses at all or your interpretation?
SP3 values fall into the range we consider to be acceptable. For all the samples, the minimum coverage was 12.07, the maximum coverage was 28.72, with a median coverage of 17.71. For SP3, the coverage of SP3_L was16.67 and SP3_R was 13.43. We are confident that an alteration in results did not occur in this sample because
- It has been determined by ENCODE that at least 10X coverage of a CpG is required for accurate measurement of percent methylation.
- We only retained sites across all samples with at least 10x coverage for downstream analyses.
- We observed the closest correlation between SP3R and SP3L.
- Line 207-210: “We found that the pairs of metachronous nonseminomas (i.e. patients SP3 and SP6) displayed a similar methylation profile but did not share any genetic mutations, suggesting a common cellular rather than genetic origin despite a separation in time of their diagnoses by 21 and 67 months, respectively.” I agree that metachronous and seminoma do not share genetic origin, but I just need a justification written about how/why you think this as a reader.
Authors: We justify our observation with the following comparison (lines 231-234): “On the other hand, even though both of patient SP2’s synchronous seminomas harbored KIT mutations, the methylation profile of his left-side seminoma more closely resembled that of patients SP23 and SP74’s right-side seminomas, which did not harbor any KIT mutation.” This is consistent with a previous observation (lines 69-71), “To date, no one has yet discovered any concordant mutations in bilateral GCT regardless of temporality (i.e. synchronous or metachronous) or histology (i.e. seminoma or nonseminoma), suggesting that no common precursor mutation has occurred during oncogenesis [11]” and that “genetic content needs to be understood in the proper cellular context” (lines 39-40, 74-75, 322-324, 447-452), which is a central theme of this paper.
- Line 233 & 237: You used adjusted p-value < 0.01 for the seminoma vs. nonseminoma and nominal p-value <0.001 for left vs. right. Why did you not just use adjusted p-value or nominal?
We have now used the adjusted p-value of < 0.01 in both comparisons.
Added to text: A similar analysis was performed to determine the differential methylation between right and left GCT in the 9 pairs with > 40% mean difference in methylation with adjusted p-value of < 0.01 and no genes were found.
Round 2
Reviewer 1 Report
Comments and Suggestions for Authors
This manuscript now can be accepted for publication in current form.
Reviewer 2 Report
Comments and Suggestions for Authors
I would like to thank the authors for their detailed response and commendable updates to the manuscript. I am completely satisfied in their updated manuscript and their thoughtful references to the comments posed on the reasoning to their paper. It clarified their logic.
I believe this work is ready for publication. Thank you for the opportunity to review your work. I thoroughly enjoyed the framing of the story and importance of your analysis. I believe the figures are very clear and look forward to your future work.